# MuSe-GNN: Learning Unified Gene Representation From Multimodal Biological Graph Data

**Tianyu Liu**
Yale University
tianyu.liu@yale.edu

**Yuge Wang**
Yale University
yuge.wang@yale.edu

**Rex Ying**
Yale University
rex.ying@yale.edu

**Hongyu Zhao***
Yale University
hongyu.zhao@yale.edu
*: Corresponding author

## Abstract

Discovering genes with similar functions across diverse biomedical contexts poses a significant challenge in gene representation learning due to data heterogeneity. In this study, we resolve this problem by introducing a novel model called Multimodal Similarity Learning Graph Neural Network, which combines Multimodal Machine Learning and Deep Graph Neural Networks to learn gene representations from single-cell sequencing and spatial transcriptomic data. Leveraging 82 training datasets from 10 tissues, three sequencing techniques, and three species, we create informative graph structures for model training and gene representations generation, while incorporating regularization with weighted similarity learning and contrastive learning to learn cross-data gene-gene relationships. This novel design ensures that we can offer gene representations containing functional similarity across different contexts in a joint space. Comprehensive benchmarking analysis shows our model's capacity to effectively capture gene function similarity across multiple modalities, outperforming state-of-the-art methods in gene representation learning by up to 97.5%. Moreover, we employ bioinformatics tools in conjunction with gene representations to uncover pathway enrichment, regulation causal networks, and functions of disease-associated or dosage-sensitive genes. Therefore, our model efficiently produces unified gene representations for the analysis of gene functions, tissue functions, diseases, and species evolution.

## 1 Introduction

Progress in biological technology has broadened the variety of biological data, facilitating the examination of intricate biological systems. A prime example of such technology is single-cell sequencing, which allows for the comprehensive characterization of genetic information within individual cells [38, 78]. This technology provides access to the full range of a cell's transcriptomics, epigenomics and proteomics information, including gene expression (scRNA-seq) [46, 114], chromatin accessibility (scATAC-seq) [20, 17], methylation [60, 68] and anti-bodies [85]. By sequencing cells from the same tissue at various time points, we can gain insight into patterns of cellular activity over time [30]. Moreover, spatial information of cellular activity represents an equally vital, additional dimension. Such data are defined as spatial data. All of these data are known as multi-omics data, and integrating multi-omics data for combined analysis poses a significant challenge.

However, the traditional idea of multi-omics data integration known as using cells as anchors [56, 87, 39, 12, 21, 76] is only partially suitable because of the following two challenges. 1. Different omics

data pose their own challenges. For example, the unit of observation of the spatial transcriptomic data is different from other single-cell data, as a single spatial location contains mixed information (see the left part of Figure 1 a) from various cells [23] and it is not appropriate to generate spatial clusters based on gene expression similarity [16]. Current research [86] also indicates that chromosome accessibility feature is not a powerful predictor for gene expression at the cell level. 2. The vast data volume in atlas-level studies challenges high-performance computing, risking out-of-memory or time-out errors [58, 101]. With nearly 37.2 trillion cells in the human body [77], comprehensive analysis is computationally infeasible. 3. Batch effects may adversely impact analysis results [54] by introducing noise. Consequently, an efficient and powerful method focusing on multi-omics and multi-tissue data (referred to as multimodal biological data) analysis is urgently needed to address these challenges.

Acknowledging the difficulties arising from the cell-oriented viewpoint, prior work shifted focus to the gene perspective. Using gene sets as a summary of expression profiles, based on natural selection during the species evolution, may provide a more robust anchor [8]. Protein-coding genes are also thought to interact with drugs [19], which are more relevant to diseases and drug discovery. Gene2vec [24] is a method inspired by Word2vec [18], which learns gene representations by generating skip-gram pairs from the co-expression network. Recently, the Gene-based data Integration and ANalysis Technique (GIANT) [16] has been developed, based on Node2vec [34] and OhmNet [119], to learn gene representations from both single-cell and spatial datasets. However, as shown in Figure 1 (b), significant functional clustering for genes from different datasets but the same tissue was not observed based on the gene embeddings from these two models, because these methods do not infer the similarity of associated genes from different multimodal data. Additionally, they did not offer metrics to quantitatively evaluate the performance of gene embeddings compared to baseline models.

Here we introduce a novel model called **Mu**ltimodal **S**imilarity **Le**arning **G**raph **N**eural **N**etwork (MuSe-GNN) [1] for multimodal biological data integration from a gene-centric perspective. The overall workflow of MuSe-GNN is depicted in Figure 1 (a). Figure 1 (b) shows MuSe-GNN's superior ability to learn functional similarity among genes across datasets by suitable model structure and novel loss function, comparing to GIANT and Gene2vec. MuSe-GNN utilizes weight-sharing Graph Neural Networks (GNNs) to encode genes from different modalities into a shared space regularized by the similarity learning strategy and the contrastive learning strategy. At the single-graph level, the design of graph neural networks ensures that MuSe-GNN can learn the neighbor information in each co-expression network, thus preserving the gene function similarity. At the cross-data level, the similarity learning strategy ensures that MuSe-GNN can integrate genes with similar functions into a common group, while the contrastive learning strategy helps distinguish genes with different functions. Furthermore, MuSe-GNN utilizes more robust co-expression networks for training and applies dimensionality reduction [89] to high-dimensional data [73].

To the best of our knowledge, this is the first paper in gene representation learning that combines the Multimodal Machine Learning (MMML) concept [9, 27] with deep GNNs [49, 81] design. Both approaches are prevalent in state-of-the-art (SOTA) machine learning research, inspiring us to apply them to the joint analysis of large-scale multimodal biological datasets [2]. As application examples, we first used the gene embeddings generated by MuSe-GNN to investigate crucial biological processes and causal networks for gene regulation in the human body. We also applied our model to analyze COVID and cancer datasets, aiming to unveil potential disease resistance mechanisms or complications based on specific differentially co-expressed genes. Lastly, we used gene embeddings from MuSe-GNN to improve the prediction accuracy for gene functions. Here genes are co-expressed means genes are connected with the same edge.

Given the lack of explicit metrics to evaluate gene embeddings, we proposed six metrics inspired by the batch effect correction problem [54] in single-cell data analysis. We evaluated our model using real-world biological datasets from one technique, and the benchmarking results demonstrated MuSe-GNN's significant improvement from **20.1%** to **97.5%** in comprehensive assessment. To summarize the advantages of our model, MuSe-GNN addresses the outlined problem about cross-data gene similarity learning and offers four major contributions: 1. Providing an effective representation learning approach for multi-structure biological data. 2. Integrating genes from different omics and tissue data into a joint space while preserving biological information. 3. Identifying co-located

---

[1]Codes of MuSe-GNN: `https://github.com/HelloWorldLTY/MuSe-GNN`
[2]Download links: Appendix M.

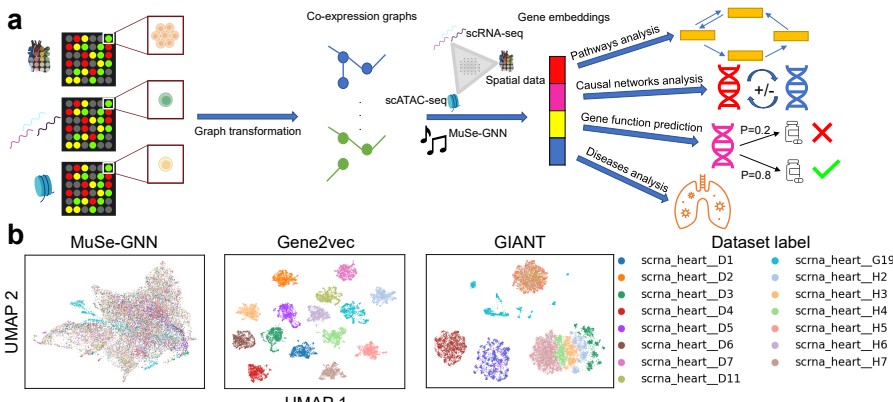

Figure 1: The workflow of MuSe-GNN, the visualization of gene embeddings, and the problems of two exitsing methods, GIANT and Gene2vec. **(a)** The process of learning gene embeddings by MuSe-GNN. Here we highlight the difference between single-cell data and spatial data, and the major applications of gene embeddings. Each spot from single-cell data represents one cell, while each spot from spatial data represents a mixture of cells. **(b)** UMAPs [63] for the gene embeddings of MuSe-GNN, Gene2vec and GIANT. They are colored by datasets. Based on UMAPs we can conclude that both Gene2vec and GIANT failed to learn the gene similarity based on the datasets from the same tissue, while MuSe-GNN produces meaningful embeddings agnostic to datasets.

genes with similar functions. 4. Inferring specialized causal networks of genes and the relationships between genes and biological pathways or between genes and diseases.

## 2 Related work

**Co-expression Network Analysis.** While direct attempts at joint analysis of gene functions across modalities are limited, there is relevant research in identifying correlation networks based on a single dataset. WGCNA [52] is a representative method that employs hierarchical clustering to identify gene modules with shared functions. However, as an early tool, WGCNA has restricted functionality. Its inference of co-expression networks necessitates more rigorous methods, and it cannot be directly applied to the analysis of multimodal biological data.

**Network Based Biological Representation Learning.** Apart from directly generating gene co-expression networks, learning quantitative gene representations in a lower dimensional space may better describe gene-gene relationships and facilitate downstream analyses. Gene2vec generates gene embeddings based on the co-expression network from a given database. However, it disregards the expression profile information and is based on the old Gene Expression Omnibus (GEO) up until 2017 [24]. GIANT leverages Node2vec [34] and OhmNet [119] to learn gene representations from both single-cell and spatial datasets by constructing graphs and hypergraphs for training. However, this approach still over-compresses multimodal biological datasets by removing expression profiles. Moreover, their co-expression networks created by Pearson correlation have a high false positive rate [82]. Additionally, some of the datasets used by GIANT are of low quality (see Appendix A). There are also methods sharing a similar objective of learning embeddings from other datasets. GSS [71] aims to learn a set of general representations for all genes from bulk-seq datasets [62] using Principal Component Analysis (PCA) and clustering analysis. However, it is for bulk-seq data and cannot be directly applied to single-cell datasets from different tissues. Gemini [103] focuses on integrating different protein function networks, with graph nodes representing proteins rather than genes.

**Graph Transformer.** In the deep learning domain, Transformer [95, 100] is one of the most successful models, leveraging seq2seq structure design, multi-head self-attention design, and position encoding design. Many researchers have sought to incorporate the advantages of Transformer to graph structure learning. TransformConv [81] introduces a multi-head attention mechanism to the graph version of supervised classification tasks and achieved significant improvements. MAGNA [98] considers higher-level hop relationships in graph data to enhance node classification capabilities. Graphomer [110] demonstrates the positive impact of the Transformer structure on various tasks

using data from the Open Graph Benchmark Large-Scale Challenge [44], which is further extended by GraphomerGD [111]. Recently, GPS [75] proposes a general Graph Transformer (GT) model by considering additional encoding types. Transformer architecture also contributes solutions to several biological questions [113]. scBERT [108] generates gene and cell embeddings using pre-training to improve the accuracy of cell type annotations. The substantial impact of these efforts highlights the crucial contribution of the Transformer architecture to graph data learning.

## 3  Methods

In the following sections of introducing MuSe-GNN, we will elaborate on our distinct approaches for graph construction that utilize multimodal biological data, followed by an explanation of our weight-sharing network architecture and the elements of the final loss function.

### 3.1  Preliminaries

**GNN.** GNNs aim to learn the graph representation of nodes (features) for data with a graph structure. Modern GNNs iteratively update the representation of a node by aggregating the representations of its $k$-order neighbors ($k \geq 1$) and combining them with the current representation. As described in [110], considering a graph $G = \{V, E\}$ with nodes $V = \{v_1, v_2, ..., v_n\}$, the AGGREGATE-COMBINE update for node $v_i$ is defined as:

$$a_i^{(l+1)} = \text{AGGREGATE}^{(l+1)} \left( \left\{ h_j^{(l)} : j \in \mathcal{N}(v_i) \right\} \right); h_i^{(l+1)} = \text{COMBINE}^{(l+1)} \left( h_i^{(l)}, a_i^{(l+1)} \right),$$
(1)

where $\mathcal{N}(v_i)$ represents the neighbors of node $v_i$ in the given graph, and $h_i^{(l)}$ and $h_i^{(l+1)}$ represent the node representation before and after updating, respectively.

**Problem Definition.** We address the gene embeddings generation task by handling multimodal biological datasets, denoted as $\mathcal{D} = (\{V_i, E_i\})_{i=1}^{T}$. Our goal is to construct a model $\mathcal{M}(\cdot, \theta)$, designed to yield gene embeddings set $\mathcal{E} = \{e_1, ..., e_T\} = \mathcal{M}(\mathcal{D}, \theta)$. In this context, $\mathcal{D}$ represents the input, $\theta$ represents the parameters, and $\mathcal{E}$ represents the output. In other words, we aim to harmonize gene information from diverse modalities within a unified projection space, thereby generating consolidated gene representations.

### 3.2  Graph construction

Before constructing gene graphs, our first contribution involves the selection of highly variable genes (HVGs) for each dataset. These HVGs constitute a group of genes with high variance that can represent the biological functions of given expression profiles. Moreover, considering co-expression networks is important for gene representation learning because it allows us to characterize gene-gene relationships. As sequencing depth, or the total counts of each cell, often serves as a confounding factor in the co-expression networks inference [11], we employ two unique methodologies, scTransform [35] and CS-CORE [88], to process scRNA-seq and scATAC-seq data, thus creating gene expression profiles and co-expression networks unaffected by sequencing depth. For spatial transcriptomic data, our focus is on genes displaying spatial expression patterns. We use SPARK-X [116] to identify such genes and then apply scTransform and CS-CORE. For detailed algorithmic information regarding these methods, please see Appendix B. Additionally, we demonstrate the immunity of CS-CORE to batch effects when estimating gene-gene correlations in Appendix C. In all our generated graphs (equivalent to co-expression datasets), nodes represent **genes** and edges represent **co-expression** relation of genes.

### 3.3  Cross-Graph Transformer

To capitalize on the strengths of the Transformer model during our training process, we integrate a graph neural network featuring a multi-head self-attention design [81], called TransformerConv, to incorporate co-expression information and generate gene embeddings. Details of TransformerConv can be found in Appendix B.4. Incorporating multimodal information can estimate more accurate gene embeddings, supported by Appendix D. The cross-graph transformer can efficiently learn gene embeddings containing gene functions across different graphs, advocated by the comparison of different network structure choices in Appendix E.1.

**Weight sharing.** Given the variability among multimodal biological datasets, we employ a weight-sharing mechanism to ensure that our model learns shared information across different graphs, representing a novel approach for learning cross-graph relationships. We also highlight the importance of weight-sharing design in Appendix E.1.

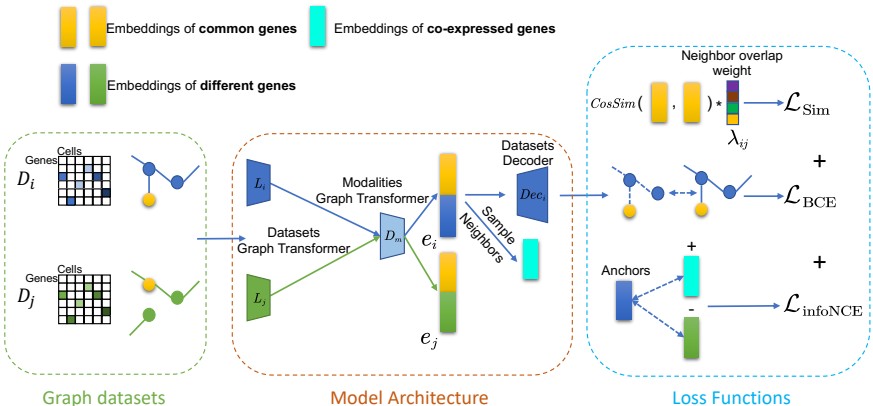

Figure 2: The overall model architecture and the design of loss functions for MuSe-GNN. The color of nodes in the green block represents common/different genes across two datasets. The brown block represents the network architecture of MuSe-GNN, and the blue block represents different loss function components of MuSe-GNN. The color gradients of the left two matrices represent different gene expression levels.

**Datasets & Modalities Graph Transformer.** Drawing inspiration from the hard weight-sharing procedure [15, 115], we not only employ dataset-specific Graph Transformer (GT) layers $L_1, L_2, ..., L_n$ for each graph $(G_1, G_2, ..., G_n)$ from the same modality $m$, but also connect all these dataset-specific layers to a set of shared GT layers, denoted as $D_m$. This design showcases our novel approach to incorporating weight-sharing into the GT framework. The forward process of MuSe-GNN, given dataset $i$ with network parameter $\theta_*$, is defined as follows:

$$X_i' = D_m(L_i(G_i; \theta_{L_i}); \theta_{D_m}). \tag{2}$$

**Datasets Decoder.** Here we propose a dataset-specific decoder structure based on Multi-layer Perceptrons (MLP). This decoder model is crucial in reconstructing the co-expression relationships among different genes, showcasing our inventive use of MLP for this purpose. Given a graph $G_i$ and its corresponding gene embedding $e_i$, the decoding process of MuSe-GNN, with network parameter $\theta_{dec,i}$, is defined as follows:

$$E_{rec} = \text{DECODERMLP}(e_i e_i^T; \theta_{dec,i}), \tag{3}$$

where $E_{rec}$ represents the reconstructed co-expression network.

### 3.4 Graph Reconstruction Loss ($\mathcal{L}_{\text{BCE}}$)

Within a single graph, we implement a loss function inspired by the Graph Auto-encoder (GAE) [48]. This function is designed to preserve two key aspects: 1. the similarity among genes sharing common functions, and 2. the distinctions among genes with differing functions. This innovative use of a GAE-inspired loss function constitutes a significant contribution to the methodological design. For a graph $G_i = \{V_i, E_i\}$, the loss function for edge reconstruction is defined as:

$$
\begin{aligned}
e_i &= \text{ENCODERGNN}(\{V_i, E_i\}; \theta_{enc}) = D_m(L_i(G_i; \theta_{L_i}); \theta_{D_m}), \\
E_{rec} &= \text{DECODERMLP}(e_i e_i^T; \theta_{dec,i}); E_{rec}' = \text{FLATTEN}(E_{rec}); E_i' = \text{FLATTEN}(E_i), \\
\mathcal{L}_{\text{BCE}} &= -\frac{1}{|E_{rec}'|} \sum_{t=1}^{|E_{rec}'|} \left[ E_i'[t] \cdot \log E_{rec}'[t] + (1 - E_i'[t]) \cdot \log(1 - E_{rec}'[t]) \right],
\end{aligned}
\tag{4}
$$

where ENCODERGNN and DECODERMLP represent the encoder and decoder parts of our model, respectively. $\mathcal{L}_{\text{BCE}}$ denotes the computation of binary cross entropy loss (BCELoss) for the given input data. $E_{rec}$ is the reconstructed adjacency matrix and $|E_{rec}'|$ is the length of its flatten version. Further justification of the model design can be found in Appendix F.

## 3.5 Weighted-similarity Learning ($\mathcal{L}_{\text{Sim}}$)

To integrate shared biological information across graphs from disparate datasets, we fuse the reconstruction loss of the input graph structure with a cosine similarity learning loss. In this process, we treat common HVGs between each pair of datasets as anchors. Our aim is to maximize the cosine similarity ($\text{CosSim}(\mathbf{a}, \mathbf{b}) = \frac{\mathbf{a} \cdot \mathbf{b}}{||\mathbf{a}||_2 \cdot ||\mathbf{b}||_2}$) for each common HVG across two arbitrary datasets (represented by the yellow blocks in Figure 2), utilizing their gene embeddings. However, in practice, different common HVGs may have different levels of functional similarity in two datasets, which is hard to be directly quantified. Consequently, we employ the shared community score as an indirect measurement, which is incorporated as a weight for the cosine similarity of different common HVG pairs within the final loss function. Considering two graphs $G_i = \{V_i, E_i\}$ and $G_j = \{V_j, E_j\}$, and one of their shared genes $g$, we identify the co-expressed genes of the given gene $g$ in both datasets, denoted as $N_{ig}$ and $N_{jg}$. Thus, the weight $\lambda_{ijg}$ for gene $g$ can be expressed as follows:

$$\lambda_{ijg} = \frac{|N_{ig} \cap N_{jg}|}{|N_{ig} \cup N_{jg}|}. \tag{5}$$

We can iterate over all shared genes from 1 to $n$ between these two graphs, ultimately yielding a vector as $\lambda_{ij} = [\lambda_{ij1}, .., \lambda_{ijn}]$. This vector encapsulates the community similarity between the two graphs across all common HVGs. We can then modify the cosine similarity of various gene pairs by first multiplying this vector with cosine similarities and then summing the resultant values across all genes. The negation of the outcome is our final weighted similarity loss, denoted as $\mathcal{L}_{\text{Sim}}$. The ablation test for weighed similarity loss can be found in Appendix E.1. More detailed explanations and evidence supporting this design in multimodal conditions can be found in Appendix G.

## 3.6 Self-supervised Graph Contrastive Learning ($\mathcal{L}_{\text{InfoNCE}}$)

Specifically, when integrating multimodal biological data, we employ the contrastive learning strategy [72] to ensure that functionally similar genes are clustered together as closely as possible, while functionally different genes are separated apart from each other. We utilize Information Noise Contrastive Estimation (InfoNCE) as a part of our loss function to maximize the mutual information between the anchor genes and genes with the same functions. This loss is applicable to different genes in two arbitrary graphs during the training process. In general, if we represent the embeddings of $N$ genes as $\text{Gene}_N = \{e_1, ..., e_N\}$, InfoNCE is designed to minimize:

$$\mathcal{L}_{\text{InfoNCE}} = -\mathbb{E}\left[\log \frac{\exp\left(e \cdot k_+ / \tau\right)}{\sum_{i=0}^{K} \exp\left(e \cdot k_i / \tau\right)}\right], \tag{6}$$

where samples $\{k_0, k_1, k_2...\}$ compose a set of gene embeddings known as keys of one dictionary and $e$ is a query gene embedding. $k_+$ represents a positive sample of $e$ and $k_i$ denotes a negative sample of $e$. Equation 6 can be interpreted as a log loss of a $(K + 1)$-way Softmax classifier, which attempts to classify $e$ as $k_+$. $\tau$ is a temperature parameter. $\tau$ is set to 0.07 referenced in MoCo [41].

## 3.7 Final Loss Function

In summary, the training objective of MuSe-GNN for graph $i$ comprises three components:

$$\min_{e_i, e_j} \mathcal{L}_{\text{BCE}}(\text{DecoderMlp}_i(e_i e_i^T), E_i) - \mathbb{E}\left[\text{CosSim}(e_i[\text{Common}_{ij}], e_j[\text{Common}_{ij}])\lambda_{ij}^T\right]$$
$$+ \lambda_c \mathcal{L}_{\text{InfoNCE}}(e_i[\text{Diff}_i] \oplus e_j[\text{Diff}_j], e_i[\text{Diff}_{\mathcal{N}(i)}] \oplus e_j[\text{Diff}_{\mathcal{N}(j)}]), \tag{7}$$

where $\text{Common}_{ij}$ denotes the index set for common HVGs, $\text{Diff}_i$ represents the index set for different HVGs in graph $i$, and $\text{Diff}_{\mathcal{N}(i)}$ indicates the index set for neighbors of $\text{Diff}_i$ in graph $i$. To expedite the training process and conserve memory usage, we sample graph $j$ for each graph $i$ during model training. We also employ multi-thread programming to accelerate the index set extraction process. $\lambda_c$ is the weight for InfoNCE loss. All of the components in MuSe-GNN are supported by ablation experiments in Appendix E.1. Details of hyper-parameter tuning can be found in Appendix E.2.

## 4 Experiments

**Datasets & Embeddings generation.** Information on the different datasets used for different experiments is included at the beginning of each paragraph. The training algorithm of our model

**Algorithm 1 Mu**ltimodal **S**imilarity **L**earning **G**raph **N**eural Network (MuSe-GNN)

---

**Input:** Model ENCODERGNN, DECODERMLP; Dataset $\mathcal{D} = (\{V_i, E_i\})_{i=1}^{T}$; Number of epochs $K$; Neighbor overlap list $\lambda$; Weight coefficient for contrastive learning $\lambda_c$;

**Helper Functions:** A function to find the genes with the same name FINDCOMMONGENES; A function to find the genes with different name FINDDIFFGENES; A function to calculate cosine similarity COSSIM; A function to sample data from a graph collection SAMPLE; A optimizer function to update weights ADAM.

**Output:** Model ENCODERGNN

1: INIT: initialize all parameters.
2: **for** $s$ in $K$ steps **do**
3:      **for** id, $\{V, E\}$ in enumerate($\mathcal{D}$) **do**
4:          $e \leftarrow$ ENCODERGNN($\{V, E\}; \theta_{enc}$); $E_{rec} \leftarrow$ DECODERMLP($ee^T; \theta_{dec}$)
5:          $\mathcal{L} \leftarrow \mathcal{L}_{\text{BCE}}(E_{rec}, E)$
6:          $\text{id}_{\text{new}}, \{V_{\text{new}}, E_{\text{new}}\} \leftarrow$ SAMPLE($G \backslash \{V, E\}$)
7:          $e_{\text{new}} \leftarrow$ ENCODERGNN($\{V_{\text{new}}, E_{\text{new}}\}; \theta_{enc}$)
8:          $e_{\text{Common}}, e'_{\text{Common}} \leftarrow$ FINDCOMMONGENES($e, e_{\text{new}}$)
9:          $\mathcal{L} \leftarrow \mathcal{L} -$ COSSIM($e_{\text{Common}}, e'_{\text{Common}}$)$\lambda_{\text{id,id}_{\text{new}}}^{T}$
10:         $e_{\text{Diff}}, e'_{\text{Diff}_N}, e_{\text{Diff}_N}, e'_{\text{Diff}_N} \leftarrow$ FINDDIFFGENES($e, e_{\text{new}}$)
11:         $\mathcal{L} \leftarrow \mathcal{L} + \lambda_c \mathcal{L}_{\text{InfoNCE}}(e_{\text{Diff}} \oplus e_{\text{Diff}_N}, e'_{\text{Diff}} \oplus e'_{\text{Diff}_N})$
12:         $\theta_{enc} \leftarrow$ ADAM($\mathcal{L}, \theta_{enc}$); $\theta_{dec} \leftarrow$ ADAM($\mathcal{L}, \theta_{dec}$)
13: Return ENCODERGNN

---

is outlined in Algorithm 1. We stored the gene embeddings in the AnnData structure provided by Scanpy [104]. To identify groups of genes with similar functions, we applied the Leiden clustering algorithm [91] to the obtained gene embeddings. Details can be found in Appendix E.4.

**Evaluation metrics.** For the benchmarking process, we used six metrics: edge AUC (AUC) [74], common genes ASW (ASW) [59], common genes graph connectivity (GC) [59], common genes iLISI (iLISI) [59], common genes ratio (CGR), and neighbors overlap (NO) to provide a comprehensive comparison. Detailed descriptions of these metrics can be found in Appendix H. We computed the metrics for the different methods and calculated the average rank (Avg Rank $\in [1, 9]$). Moreover, to evaluate the model performance improvement, we applied min-max scaling to every metric across different models and computed the average score (Avg Score $\in [0, 1]$).

**Baselines.** We selected eight models as competitors for MuSe-GNN. The first group of methods stems from previous work on learning embeddings for biomedical data, including Principal Component Analysis (PCA) [74] used by GSS, Gene2vec, GIANT (the SOTA model for gene representation learning), Weight-sharing Multi-layer Auto-encoder (WSMAE) used by [109] and scBERT (the SOTA model with pre-training for cell type annotation). The second group of methods comprises common unsupervised learning baseline models, including GAE, VGAE [48] and Masked Auto-encoder (MAE) (the SOTA model for self-supervised learning) [42]. MuSe-GNN, GIANT, GAE, VGAE, WSMAE and MAE have training parameter sizes between 52.5 and 349 M, and all models are tuned to their best performance. Details are shown in Appendix E.2.

**Biological applications.** For the pathway analysis, we used Gene Ontology Enrichment Analysis (GOEA) [1] to identify specific biological pathways enriched in distinct gene clusters with common functions. Moreover, we used Ingenuity Pathway Analysis (IPA) [2] to extract biological information from the genes within various clusters, including causal networks [50] and sets of diseases and biological functions. Biological pathways refer to processes identified based on the co-occurrence of genes within a particular cluster. The causal network depicts the relationships between regulatory genes and their target genes. Disease and biological function sets facilitate the discovery of key processes and complications associated with specific diseases. Using gene embeddings also improves the performance of models for gene function prediction. To visualize gene embeddings in a low-dimensional space, we utilized Uniform Manifold Approximation and Projection (UMAP) [63].

Table 1: Avg Score across different tissues. Standard deviations are reported in Appendix E.3.

| Methods | Heart | Lung | Liver | Kidney | Thymus | Spleen | Pancreas | Cerebrum | Cerebellum | PBMC |
|---|---|---|---|---|---|---|---|---|---|---|
| PCA | 0.52 | 0.48 | 0.56 | 0.47 | 0.56 | 0.60 | 0.51 | 0.62 | 0.53 | 0.51 |
| Gene2vec | 0.40 | 0.37 | 0.33 | 0.29 | 0.21 | 0.31 | 0.24 | 0.27 | 0.31 | 0.19 |
| GIANT | 0.50 | 0.40 | 0.33 | 0.38 | 0.58 | 0.33 | 0.56 | 0.29 | 0.28 | 0.28 |
| WSMAE | 0.50 | 0.47 | 0.54 | 0.46 | 0.57 | 0.53 | 0.52 | 0.55 | 0.59 | 0.50 |
| GAE | 0.61 | 0.45 | 0.58 | 0.40 | 0.56 | 0.58 | 0.52 | 0.56 | 0.60 | 0.54 |
| VGAE | 0.64 | 0.32 | 0.33 | 0.38 | 0.56 | 0.31 | 0.33 | 0.41 | 0.33 | 0.47 |
| MAE | 0.36 | 0.47 | 0.50 | 0.45 | 0.41 | 0.52 | 0.39 | 0.50 | 0.49 | 0.50 |
| scBERT | 0.41 | 0.49 | 0.55 | 0.62 | 0.17 | 0.58 | 0.46 | 0.60 | 0.61 | 0.58 |
| MuSeGNN | **0.77** | **0.96** | **0.92** | **0.89** | **0.89** | **0.94** | **0.80** | **0.95** | **0.90** | **0.92** |

## 4.1 Benchmarking Analysis

We executed each method 10 times by using the same setting of seeds to show the statistical significance based on datasets across different tissues. The performance comparison of nine gene embedding methods is presented in Tables 1 and 14. Based on these two tables, MuSe-GNN outperformed its competitors in terms of both average ranks and average scores across all the tissues. Based on Table 1, for major tissues, such as heart and lung, MuSe-GNN's performance was 20.1% higher than the second-best method and 97.5% higher than the second-best method in heart and lung tissue, respectively. According to Appendix E.3, MuSe-GNN's stability was also demonstrated through various metrics by comparing standard deviations, including AUC, GC, and NO. In contrast, methods such as Gene2vec, GAE, VGAE, MAE and scBERT exhibited significant instability in their evaluation results for kidney or thymus. Consequently, we concluded that MuSe-GNN is the best performing model for learning gene representation based on datasets from different tissues, making it applicable to learn gene embeddings from diverse multimodal biological data.

## 4.2 Analysis of Gene Embeddings from Multimodal Biological Data

In Figure 3, we displayed the integration results for multimodal biological data from Humans. Figure 3 (a) and (b) demonstrated that MuSe-GNN could successfully integrate genes from different modalities into a co-embedded space, allowing us to identify functional groups using the Leiden algorithm shown in Figure 3 (c). Furthermore, Figure 3 (d) revealed that most of the clusters in (c) were shared across different modalities. We also identified three significant functional groups in Figure 3 (a): the nervous system (predominantly composed of the cerebrum and cerebellum [26]), the cardiovascular system (mainly composed of heart, lung, and kidney [29]), and the immunology system (primarily consisting of spleen, liver, and peripheral blood mononuclear cells (PBMC) [70]). All systems are important in regulating the life activities of the body. We also uncovered a pre-epigenetics group (mainly consisting of scATAC-seq data without imprinting, modification, or editing), emphasizing the biological gap existing in multi-omics and the importance of post-transcriptional regulation.

Using GOEA, we could identify significant pathways enriched by different co-embedded gene clusters. For instance, Figure 3 (d) displayed the top 5 pathways in an immunology system cluster. The rank was calculated based on the negative logarithm of the false discovery rate. Since all top pathways were related to immunological defense and response, it further supported the accuracy of our embeddings in representing gene functions. For our analysis of shared transcription factors and major pathways across all the tissues, please refer to Appendix I. For our analysis of multi-species gene embeddings, please refer to Appendix J.

## 4.3 Analysis of Gene Embeddings for Diseases

We generated gene embeddings for human pancreas cells from samples with and without SARS-CoV-2 infection, as depicted in Figure 4 (a) and (b). We identified specific genes from COVID samples that did not align with control samples, which piqued our interest. These genes, highlighted by a red circle in Figure 4 (c), could be interpreted as differentially functional genes in diseased cells.

We conducted GOEA for the genes of interest and discovered a close relationship among these gene enrichment results and the top 5 pathways associated with immune activity. These results are displayed in Figure 4 (d). For the genes within our target cluster, we utilized IPA to identify the Entrez name of these genes, and 90.3% (122/135) genes in our cluster are related to immunoglobulin.

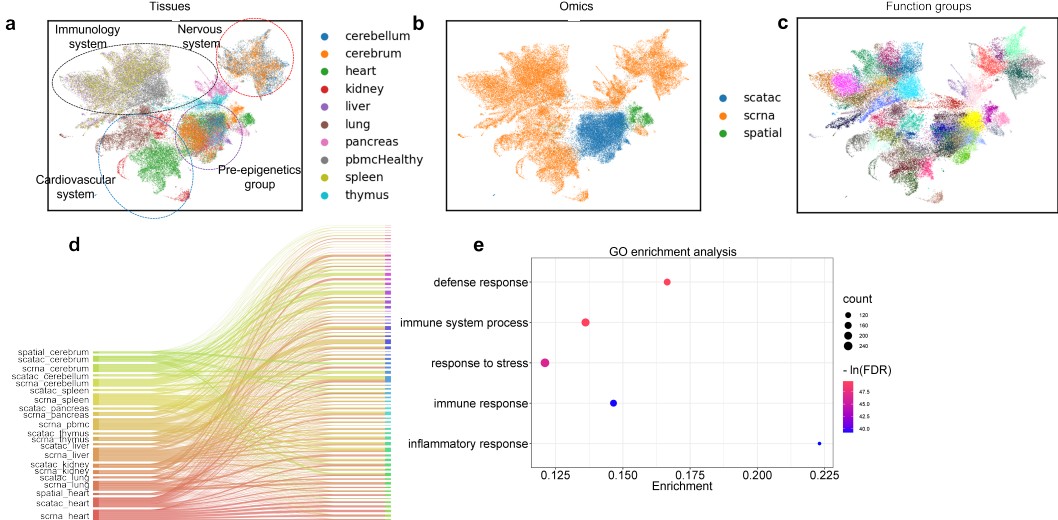

Figure 3: Gene representation learning results for multimodal biological data. **(a)** represents the UMAPs of gene embeddings colored by tissue type and highlighted by biological system. **(b)** represents the UMAPs of gene embeddings colored by omics type. **(c)** represents the UMAPs of gene embeddings colored by common function groups. **(d)** is a Sankey plot [80] to show the overlap of different modalities in the same clusters. **(e)** shows the top5 pathways related to the genes in the special cluster discovered by GOEA. The bubble plots in this paper were created based on ggplot2 [102].

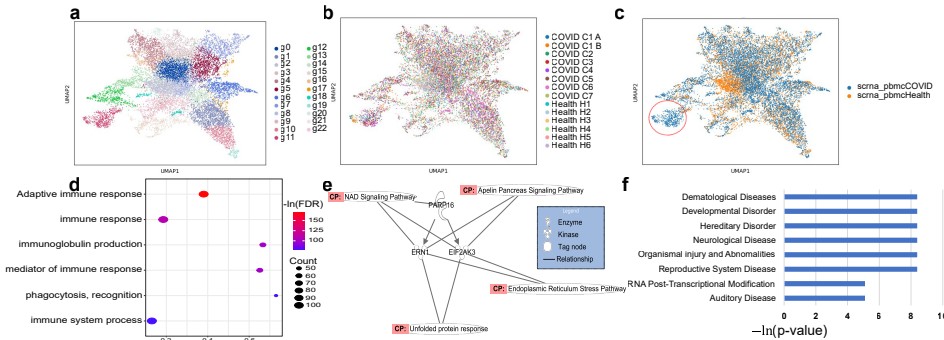

Figure 4: Gene embeddings from COVID samples and healthy samples. **(a)** represents the UMAPs of gene embeddings colored by functional groups. **(b)** represents the UMAPs of gene embeddings colored by datasets. **(c)** represents the gene embeddings colored by the conditions, and the red circle reflects the differential co-expression genes. **(d)** shows the top6 pathways related to the genes in the special cluster discovered by GOEA. **(e)** represents the causal network existing in the special cluster discovered by IPA. **(f)** represents the top diseases & biological functions discovered by IPA.

We could also infer the causal relationship existing in the gene regulatory activity of the immune system. For example, Figure 4 (e) showed a causal network inferred by IPA based on our genes cluster. PARP16, as an enzyme, can regulate ERN1 and EIF2AK3, and certain pathways are also related to this causal network. Moreover, we also showed the relation between the set of genes and Disease & Bio functions in Figure 4 (f). We identified top related Diseases & Bio functions ranked by negative logarithm of p-value, and all of these diseases could be interpreted as complications that may arise from new coronavirus infection [6, 64, 79, 69, 33, 7, 83, 84]. Our extra analyses for lung cancer data can be found in Appendix K.

### 4.4 Analysis of Gene Embeddings for Gene Function Prediction.

Here we intend to predict the dosage-sensitivity of genes related to genetic diagnosis (as dosage-sensitive or not) [90]. We used MuSe-GNN to generate gene embeddings for different datasets based on an unsupervised learning framework and utilized the gene embeddings as training dataset to predict the function of genes based on k-NN classifier. k-NN classifier is a very naive model and can reflect the contribution of gene embeddings in the prediction task.

In this task, we evaluated the performance of MuSe-GNN based on the dataset used in Geneformer [90, 31], comparing it to the prediction results based on raw data or Geneformer (total supervised learning). As shown in Table 2, the prediction accuracy based on gene emebddings from MuSe-GNN is the highest one. Moreover, the performance of gene embeddings from MuSe-GNN is better than Geneformer, which is a totally supervised learning model. Such finding proves the advantages of MuSe-GNN in the application of gene function prediction task. Further application analysis can be found in Appendix L.

Table 2: Accuracy for dosage-sensitivity prediction

|  | **MuSe-GNN (unsup)** | **Geneformer (sup)** | **Raw** |
|---|---|---|---|
| Accuracy | 0.77±0.01 | 0.74±0.06 | 0.75±0.01 |

## 5 Conclusion

In this paper, we introduce MuSe-GNN, a model based on Multimodal Machine Learning and Deep Graph Neural Networks, for learning gene embeddings from multi-context sequencing profiles. Through experiments on various multimodal biological datasets, we demonstrate that MuSe-GNN outperforms current gene embedding learning models across different metrics and can effectively learn the functional similarity of genes across tissues and techniques. Moreover, we performed various biological analyses using the learned gene embeddings, leveraging the capabilities of GOEA and IPA, such as identifying significant pathways, detecting diseases, and inferring causal networks. Our model can also contribute to the study of the pathogenic mechanisms of diseases like COVID and lung cancer, and improve the prediction performance for gene functions. Overall, the gene representations learned by MuSe-GNN are highly versatile and can be applied to different analysis frameworks.

At present, MuSe-GNN does not accept graphs with nodes other than genes as input. In the future, we plan to explore more efficient approaches for training large models related to Multimodal Machine Learning and extend MuSe-GNN to a more general version capable of handling a broader range of multimodal biological data.

## 6 Acknowledgements

This research was supported in part by NIH R01 GM134005, R56 AG074015, and NSF grant DMS 1902903 to H.Z. We appreciate the comments, feedback, and suggestions from Chang Su, Zichun Xu, Xinning Shan, Yuhan Xie, Mingze Dong, and Maria Brbic.

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
