# OpenReview forum: "MuSe-GNN: Learning Unified Gene Representation From Multimodal Biological Graph Data"
_NeurIPS.cc/2023/Conference — NeurIPS 2023 poster_

### Official Review · Reviewer_vvZM · 2023-06-28

**Soundness:** 4 excellent
**Presentation:** 4 excellent
**Contribution:** 4 excellent
**Rating:** 8
**Confidence:** 4

**Summary:**

This paper introduces MuSe-GNN, a model for learning gene embeddings from single-cell sequencing and spatial transcriptomic data that is based on multimodal machine learning and deep graph neural networks.  While incorporating regularization with weighted similarity learning and contrastive learning to learn cross-data gene-gene relationships, the proposed method creates informative graph structures for model training and the generation of gene representations, which can make the learned gene representations contain functional similarity across various contexts in a joint space. The results of tests on diverse multimodal biological datasets show that MuSe-GNN can efficiently learn the functional similarity of genes across tissues and methods and surpasses current gene embedding learning models across several metrics. This model can aid in the investigation of the pathogenic processes underlying conditions like COVID and lung cancer.

**Strengths:**

1. The proposed model learns the gene expression, which incorporates the regularization with weighted similarity learning and contrastive learning to learn cross-data gene-gene relationships. The proposed model can analyze the large-scale multimodal biological datasets.
2. The model is applied to analyze COVID and cancer datasets, which helps to unveil potential disease resistance mechanisms or complications based on specific differentially co-expressed genes.
3. MuSe-GNN has obtained significant improvement from 24.2% to 100.4% in comprehensive assessment.
4. The overall pipeline and the mechanisms are presented in detail.

**Weaknesses:**

1. What is the difference between this paper and the Geneformer [1]. What about the comparisons of the performance of the experiments if they can be compared together?
2. In Figure 1, why Gene2vec and GIANT failed to learn the gene similarity? The embeddings are clustered by datasets.

[1] Theodoris, Christina V., Ling Xiao, Anant Chopra, Mark D. Chaffin, Zeina R. Al Sayed, Matthew C. Hill, Helene Mantineo et al. "Transfer learning enables predictions in network biology." Nature (2023): 1-9.

**Questions:**

See above.

**Limitations:**

Yes, the authors have discussed their limitations and potential impact of gene embeddings for diseases, COVID.

---

> ### Author Rebuttal · Authors · 2023-08-08
>
> We thank the reviewer for the supportive comments. The detailed response to each point is as follows.
>
> **1. What is the difference between this paper and the Geneformer [1]. What about the comparisons of the performance of the experiments if they can be compared together?**
>
> We appreciate your questions.  The distinguishing features between the work presented in this paper, namely MuSe-GNN, and Geneformer can be summarized as follows:
>
> 1. Geneformer is a Large Language Model (LLM) for single-cell data, which leverages a large-scale single-cell RNA-seq (scRNA-seq) dataset (a single modality). It employs a self-supervised pre-training approach, similar to BERT [2], to predict cell types or gene functions through fine-tuning on specific datasets.
>
> 2. MuSe-GNN is specifically designed for gene representation learning, leveraging large-scale single-cell multi-omics data to create meaningful gene embeddings through an unsupervised learning framework.
>
> The main function of Geneformer is to perform annotation and prediction, while it does not guarantee the generation of high-quality embeddings. The training framework of Geneformer is supervised and needs labels, in contrast to our unsupervised approach. Geneformer also has more limited application scenarios based on our following experiments. Moreover, we have already benchmarked a similar method, known as scBERT [3]. Therefore, MuSe-GNN has more functions than Geneformer, and can handle multimodal datasets. Moreover, MuSe-GNN is not as resource-intensive and time-consuming (the limitation of LLM) as Geneformer.
>
> Regarding possible shared experiments between Geneformer and MuSe-GNN, we considered both the gene function prediction task, a key feature of Geneformer, and the gene representation learning task, central to MuSe-GNN.
>
> For the first task, we produced gene embeddings from the dataset utilized in [4] via MuSe-GNN and employed Support Vector Machine [5] as the classifier. This dataset was previously used by Geneformer for the gene function prediction task. The resulting accuracy scores are presented in Table 1.
>
> Table 1: Accuracy of gene function prediction task. Here "unsup" means unsupervised learning and "sup" means supervised learning.
>
> |          | MuSe-GNN (unsup) | Geneformer (sup) | Raw          |
> |----------|------------------|------------------|--------------|
> | Accuracy | 0.77 $\pm$ 0.01     | 0.74 $\pm$ 0.06     | 0.75 $\pm$ 0.01 |
>
> We present both the average accuracy and standard deviation for three different configurations: one based on gene embeddings derived from MuSe-GNN, another based on Geneformer, and the last one based on raw gene expression data (Raw). The classification process for raw data also utilizes the Support Vector Machine.
>
> Our results indicate that MuSe-GNN's performance is comparable to that of Geneformer, yet with less variability, suggesting that MuSe-GNN is more robust. Importantly, while Geneformer operates on the principles of supervised learning, MuSe-GNN employs an unsupervised approach. As a result, the gene embeddings produced by MuSe-GNN are capable of being utilized across a broader range of downstream tasks, as they encapsulate more meaningful biological contexts.
>
> For the second task, we attempted to generate gene embeddings using Geneformer and record its performance. However, due to the absence of gene names in Geneformer's input and output, the benchmarking process could not be executed. Moreover, the absence of gene names greatly affects downstream applications; for example, analyses like pathway enrichment or disease analysis become infeasible. Hence, MuSe-GNN has more functions than Geneformer.
>
> **2. In Figure 1, why Gene2vec and GIANT failed to learn the gene similarity? The embeddings are clustered by datasets.**
>
> We appreciate your insightful observation. We have the following explanation:
>
> This observed phenomenon arises due to these methods failing to accurately capture similar associated genes across various multimodal data. More details can be found in the Introduction section of our manuscript.
>
> The UMAP visualizations of GIANT's gene embeddings reveal that it only learned such information for parts of datasets. Therefore, GIANT does not effectively capture the common gene information that spans different datasets and contexts. The UMAP visualizations of Gene2vec reveal that it failed in learning such information across all the datasets.
>
> In contrast, our proposed model, MuSe-GNN, addresses this issue through the integration of a cross-graph Transformer model along with various loss functions. This unique approach enables our model to efficiently capture and assimilate information from multiple datasets. This results in improved gene similarity learning and effectively overcomes the dataset bias that both Gene2vec and GIANT encounter. Our design for such a function was explained in the Methods section and Appendix D.1 of the Supplementary Materials.
>
> Please let us know if you have further questions, and we will do our best to address them.  Once again, we thank you for your thoughtful review.
>
> References:
>
> [1] Theodoris, Christina V., Ling Xiao, Anant Chopra, Mark D. Chaffin, Zeina R. Al Sayed, Matthew C. Hill, Helene Mantineo et al. "Transfer learning enables predictions in network biology." Nature (2023): 1-9.
>
> [2] Devlin, Jacob, et al. "Bert: Pre-training of deep bidirectional transformers for language understanding." arXiv preprint arXiv:1810.04805 (2018).
>
> [3] Yang, Fan, et al. "scBERT as a large-scale pretrained deep language model for cell type annotation of single-cell RNA-seq data." Nature Machine Intelligence 4.10 (2022): 852-866.
>
> [4] Franzén, Oscar, Li-Ming Gan, and Johan LM Björkegren. "PanglaoDB: a web server for exploration of mouse and human single-cell RNA sequencing data." Database 2019 (2019): baz046.
>
> [5] Pedregosa, Fabian, et al. "Scikit-learn: Machine learning in Python." the Journal of machine Learning research 12 (2011): 2825-2830.

---

> > ### Comment · Reviewer_vvZM · 2023-08-16
> >
> > I have read your rebuttal and appreciate your effort. It is good to compare with Geneformer. Good luck!

---

> > > ### Author Response · Authors · 2023-08-16
> > > **Thanks!**
> > >
> > > Thank you very much for your recognition and wishes! Your time and efforts are greatly appreciated.

---

### Official Review · Reviewer_mFBG · 2023-07-06

**Soundness:** 3 good
**Presentation:** 3 good
**Contribution:** 3 good
**Rating:** 6
**Confidence:** 4

**Summary:**

The authors proposed a novel deep graph neural network, named MuSe-GNN, to learn gene representations from single-cell sequencing and spatial transcriptomic data.
The idea is to construct gene graphs from single-cell sequencing and spatial transcriptomic data, and then learn gene embeddings via cross-graph transformer.

**Strengths:**

The tasks of learning gene embeddings from multi-context sequencing profiles is of interest especially in the bioinformatic domain.
The empirical results seem to indicate that the approach can be applied to different analysis frameworks and the gene representations learned by their approach are highly versatile.
Writing and presentation skill is well.

**Weaknesses:**

This idea is not new, and the models they used are all well established.
Lack of comparison with related works.
Lack of ablation studies.

**Questions:**

One additional ablation experiment would be of interest: how much would the performance be impacted if you induce the embedding using only part of the modules.
They can compare their method with other deep learning-based multimodal representation learning models, such as [1].
[1] Lin X, Tian T, Wei Z, et al. Clustering of single-cell multi-omics data with a multimodal deep learning method[J]. Nature communications, 2022, 13(1): 7705.

**Limitations:**

N/A.

---

> ### Author Rebuttal · Authors · 2023-08-08
>
> We thank the reviewer for the supportive comments. The detailed response to each point is as follows.
>
> **1.   This idea is not new, and the models they used are all well established. Lack of comparison with related works. Lack of ablation studies.**
>
> We appreciate your comments concerning the novelty of our work, which we address in two distinct aspects.
>
> The first aspect pertains to the novelty of our model objective. As outlined in the Introduction section, the conventional method of multi-omics data integration hinges on using cells as dimensionality reduction anchors. Such approach, however, faces significant computational hurdles including data-specific challenges, substantial data volumes, and batch effects.
>
> Therefore, we propose to shift our focus to gene perspective. The existing models fail in integrating gene information across different multimodal biological data. By designing MuSe-GNN, we aim to address a fundamental constraint of a representation learning problem in the computational biology area. We provide more details in the Introduction section.
>
> The second aspect revolves around the novelty of our methodology framework. To our knowledge, our work is the first instance in gene representation learning where the Multimodal Machine Learning (MMML) paradigm is synergistically paired with deep Graph Neural Network (GNN) designs. This combination has enabled us to achieve state-of-the-art performance according to our experimental results.
>
> Furthermore, our core model comprises three unique components, each validated through an ablation study presented in Appendix D.1 of the Supplementary Materials. Finally, we consider constructing our graph data based on **multimodal data context** and **statistical test**, which is also novel and reliable.
>
> Regarding the question about the model established level, we think that the **Cross-graph Transformer** design remains an active area of exploration, especially in the computational biology track. Furthermore, we introduced the **Weighted-Similarity Learning** design, which represents a novel attempt to enhance the performance of Graph Neural Networks (GNN). Please find more details about these design elements in the Methods section of our manuscript.
>
> Regarding the question about model comparisons, our analysis involves a range of competitors that span gene representation learning methods (Gene2vec, GIANT), representative Graph Neural Networks (GNNs) (GAE, VGAE), and methods rooted in pre-training (scBERT) or other machine learning frameworks (PCA, WSMAE, MAE). We also intend to include Geneformer [1], a relatively recent model, in our comparisons. As indicated in Table 1 of our main text, our model outperformed all these competitor methods. We think our benchmarking is comprehensive, as noted by Reviewers cB5G and vvZM.
>
> Regarding the question about ablation studies, please see the information from Appendix D.1 of the Supplementary Materials, where we tried the ablation tests with different loss functions, different input settings, the contribution of cross-graph design, and the performance of different GNN structures. As noted by reviewer cB5G, our ablation tests are exhaustive and informative.
>
> **2.   One additional ablation experiment would be of interest: how much would the performance be impacted if you induce the embedding using only part of the modules. They can compare their method with other deep learning-based multimodal representation learning models, such as [2].**
>
> We appreciate your suggestions. Regarding the possible ablation test about dropping modalities (We assume 'modules' was meant to refer to 'modalities'; please let us know if this is not the case), we would like to emphasize the crucial role that data from diverse contexts play in our model's design. Comparisons between data with a greater number of modalities and those with fewer might be deemed unfair due to the variations in the sizes of the training datasets.
>
> Additionally, theoretical research, as exemplified by [3], has demonstrated that the introduction of more modalities can enhance a model's performance in addressing related downstream tasks. We extrapolate this principle to the domain of gene representation learning. The central premise here is that the inclusion of data from a wider range of modalities can generate a mapping function that features a reduced upper bound of loss. Such information will be included in our updated version.
>
> Regarding the comparison with [2], we posit that such a comparison might not be entirely relevant due to two significant differences between the methods. Firstly, scMDC is designed for cell embeddings, while our model centers around gene embeddings. Secondly, scMDC necessitates 'paired' scRNA-ADT or scRNA-scATAC data, which are not only limited in availability but also differ considerably from the input our model requires. Instead of a direct comparison with scMDC, we have expanded on the comparison between MuSe-GNN and Geneformer in our rebuttal suggested by Reviewer vvZM.
>
> Nevertheless, we will discuss this paper [2] as an example of learning cell embeddings in our Related work section.
>
> Please let us know if you have any further questions, and we will try our best to address them. Once again, we thank you for your thoughtful reviews.
>
> References:
>
> [1] Theodoris, Christina V., et al. "Transfer learning enables predictions in network biology." Nature (2023): 1-9.
>
> [2] Lin X, Tian T, Wei Z, et al. Clustering of single-cell multi-omics data with a multimodal deep learning method[J]. Nature communications, 2022, 13(1): 7705.
>
> [3] Huang, Yu, et al. "What makes multi-modal learning better than single (provably)." Advances in Neural Information Processing Systems 34 (2021): 10944-10956.

---

### Official Review · Reviewer_anSh · 2023-07-06

**Soundness:** 3 good
**Presentation:** 2 fair
**Contribution:** 3 good
**Rating:** 6
**Confidence:** 4

**Summary:**

This paper addresses the heterogeneity problem in the gene representation learning from multi-context biomedical sequencing profiles. Specifically, gene connections are formulated through co-expression network, then the proposed MuSe-GNN utilizes cross-graph Transformer to generate gene embeddings, while designing a contrastive-learning based regularization term to integrate multi-modal data.

**Strengths:**

1. The paper is well-motivated. Effective gene representation learning from multi-context profile datasets is critical real-world biomedical applications.
2. The experimental results demonstrate that MuSe-GNN achieves the state-of-art performance and brought a significant improvement.
3. Bioinformatical analysis on identifying significant pathways, diseases, and causal networks provides biologically-informed insights .

**Weaknesses:**

1. The novelty of the proposed method is limited.
2. The methodology part is not adequately described. The cross-graph Transformer section is hard to follow and Figure 2 can be improved.

**Questions:**

Minors:
The application area is rather specialized, so this paper may be of interest to a smaller subset of the NeurIPS community.

**Limitations:**

Yes.

---

> ### Author Rebuttal · Authors · 2023-08-08
>
> We thank the reviewer for the supportive comments. The detailed response to each point is as follows.
>
>
> **1.   The novelty of the proposed method is limited.**
>
> We appreciate your inquiry concerning the novelty of our work, which we address in the following from two aspects.
>
> The first aspect pertains to the novelty of our model objective. As outlined in the Introduction section, the conventional method of multi-omics data integration hinges on using cells as dimensionality reduction anchors. Such approach, however, faces significant computational hurdles including data-specific challenges, substantial data volumes, and batch effects.
>
> Therefore, we propose to shift our focus to gene perspective. The existing models fail in integrating gene information across different multimodal biological data. By designing MuSe-GNN, we aim to address a fundamental constraint of a representation learning problem in the computational biology area. We provide more details in the Introduction section.
>
> The second aspect revolves around the novelty of our methodology framework. To our knowledge, our work is the first instance in gene representation learning where the Multimodal Machine Learning (MMML) paradigm is synergistically paired with deep Graph Neural Network (GNN) designs. This combination has enabled us to achieve state-of-the-art performance according to our experimental results.
>
> Furthermore, our core model comprises three unique components, each validated through an ablation study presented in Appendix D.1 of the Supplementary Materials. Finally, we consider constructing our graph data based on **multimodal data context** and **statistical test**, which is also novel and reliable.
>
> **2.   The methodology part is not adequately described. The cross-graph Transformer section is hard to follow and Figure 2 can be improved.**
>
> We appreciate your comment on the lack of clarity of the description of our method. For the explanation of graph transformer design, we can add more details as follows:
>
> Regarding the description of our methodology, we have summarized all the essential components in the Methods section of our main text. Due to the page limit, we extended our method explanation in Appendix B of the Supplementary Materials.
>
> Regarding our cross-graph transformer design, we have addressed it in the main text and appendix. We offered comprehensive explanation in Section 3.3 of the Methods in our manuscript. Additionally, we have provided more details about our graph transformer architecture in Appendix B.4 of the Supplementary Materials.
>
> We also improved Figure 2 with more annotations and attached it to the Author Rebuttal according to the rebuttal requirement. Please let us know if you have further questions or comments.
>
> The updated figure description is: The overall model architecture and the design of loss functions for MuSe-GNN. The brown block represents the network architecture of MuSe-GNN. The blue block represents different loss function components of MuSe-GNN. The green block represents the input datasets. The color gradients of the left two matrices represent different gene expression levels. All the components of MuSe-GNN and gene embeddings are annotated.
>
> **3. This paper may be of interest to a smaller subset of the NeurIPS community.**
>
> We appreciate your comment. As for the concerns regarding the specialized nature of our topic, we respectfully offer a different perspective. The intersection of AI and science [1], particularly within the realm of biology, has been an integral part of top-tier machine learning conferences in the past decades. The application of AI technology to address critical biological questions holds immense excitement and significance. This is evidenced by groundbreaking works such as Alphafold2 [2], which revolutionized protein design.
>
> We note that there are many papers and activities about computational biology in NeurIPS, including:
>
> **Papers:**
>
> Hetzel, Leon, et al. "Predicting cellular responses to novel drug perturbations at a single-cell resolution." Advances in Neural Information Processing Systems 35 (2022): 26711-26722.
>
> Xiao, Feiyi, et al. "Estimating graphical models for count data with applications to single-cell gene network." Advances in Neural Information Processing Systems 35 (2022): 29038-29050.
>
> Tu, Xinming, et al. "Cross-linked unified embedding for cross-modality representation learning." Advances in Neural Information Processing Systems 35 (2022): 15942-15955.
>
> **Workshops:**
>
> LMRL - Learning Meaningful Representations of Life (https://www.lmrl.org/).
>
> AI for Science (https://ai4sciencecommunity.github.io/neurips23/).
>
> Machine Learning in Structural Biology Workshop (https://neurips.cc/virtual/2023/workshop/66513).
>
> **Competitions:**
>
> Open Problems in Single Cell Analysis - NeurIPS 2022: Multimodal Single-Cell Integration Across Time, Individuals, and Batches (https://openproblems.bio/events/2022-08_neurips/) (More than 1000 teams).
>
> Open Problems in Single Cell Analysis - Coming August 2023 - Single-Cell Perturbation Prediction (https://openproblems.bio/events/2023-08_neurips/).
>
> We think these references underscore the impact and relevance of our work within the realm of AI for science and biology. As commented by reviewer vvZM, our manuscript has excellent impact on at least one area, or high-to-excellent impact on multiple areas. Therefore, we think our research will capture the interest of scholars beyond these disciplines. There is also likely knowledge transfer, as a key property of machine learning research, among different areas.
>
> Please let us know if you have further questions, and we will do our best to address them. Once again, we thank you for your thoughtful review.
>
> References:
>
> [1] Wang, Hanchen, et al. "Scientific discovery in the age of artificial intelligence." Nature 620.7972 (2023): 47-60.
>
> [2] Jumper, John, et al. "Highly accurate protein structure prediction with AlphaFold." Nature 596.7873 (2021): 583-589.

---

> > ### Comment · Reviewer_anSh · 2023-08-19
> >
> > I have read your rebuttal and  my concerns are well addressed. Thus, I have increased my score.

---

> > > ### Author Response · Authors · 2023-08-19
> > > **Thanks a lot!**
> > >
> > > Thank you very much for your recognition. Your time and efforts are greatly appreciated.

---

### Official Review · Reviewer_cB5G · 2023-07-07

**Soundness:** 2 fair
**Presentation:** 3 good
**Contribution:** 3 good
**Rating:** 7
**Confidence:** 4

**Summary:**

The paper describes an approach that combines Multimodal Machine Learning and Deep Graph Neural Networks to learn gene representations from multi-omics and multi-tissue data. The main issue that this paper tries to address is that existing approaches fail to obtain gene representations that are consistent across modalities (esp. from different data sources but same tissue). To accomplish this goal, the authors propose a model called MuSE-GNN that takes as input graph co-expression networks extracted from different modalities and projects them into a unified and consistent latent space to extract gene embeddings. These gene embeddings are then used for several downstream tasks such as pathway analysis, causal network analysis, and disease analysis. In particular, the model is learned through a loss function composed of three terms: a co-expression graph reconstruction term, a weighted similarity term (where the pairwise similarity of gene representations across modalities is maximized), and a self-supervised contrastive term (to cluster together functionally similar genes and pull apart functionally different genes).

The model is evaluated on a benchmark consisting of different tissues against 9 other methods, where it shows excellent performance on different metrics. In a subsequent experiment, the authors show that MuSE-GNN embeddings successfully relate to functional groups in human data. Lastly, they present a case study on COVID-19 where they run the model on pancreatic human cells of healty vs diseased COVID patients to identify differentially expressed genes, showing (with gene enrichment) that these genes are related to immune system activity and that some of them are part of known causal networks of the regulatory activity of the immune system.

**Strengths:**

This paper is well-written and easy to follow. It studies a relevant problem (how to obtain consistent gene embeddings across modalities) and proposes a quite elegant solution (make them consistent mainly through a tailored loss function). The experimental results are good and the ablations (in the appendix) are exhaustive and very informative.

**Weaknesses:**

It seems that the competitors have been evaluated with default hyper-parameters, while the hyper-parameters of MuSE-GNN have been tuned. This is unfair, since the results of MuSE-GNN are optimized to the experimental settings of this paper, while those of the competitors are not. This might be addressed by tuning (some of) the hyper-parameters of the competitors, although I understand it might be difficult given the time constraints of the review process.

**Questions:**

- from the code, I can see that the competitors are trained with random seeds in the range [0, 10) while I can't see the same for MuSE-GNN. Can you confirm that MuSE-GNN has been trained with the same random seeds and thus the results are comparable?
- can you explain better this sentence in the appendix: "Methods such as WSMAE, GAE, VGAE, and MAE, which are based on MLP/GNNs and inspired by other methods, are configured with the best hyperparameters based on parameter searching to ensure fairness."?
- Your model uses 349M parameters. How many parameters GIANT uses? If they don't match, can you train a version of MuSE-GNN with a comparable number of parameters to GIANT (or alternatively, a version of GIANT with a number of parameters comparable to MuSE-GNN)?

**Limitations:**

Not discussed. I don't see many limitations of this approach besides getting data of proper quality (which is independent of the contribution). Perhaps, as mentioned by the authors, the model is quite "heavy" in terms of number of parameter and one needs to bring down the embedding dimension to avoid OOM errors.

No negative societal impact of concern.

---

> ### Author Rebuttal · Authors · 2023-08-08
>
> We thank the reviewer for the supportive comments. The detailed response to each point is as follows.
>
> **1. It seems that only the hyper-parameters of MuSe-GNN were tuned, while other competitors were not.**
>
> We appreciate your question. We have mentioned the details of parameter tuning for other methods in Appendix D.2 of the Supplementary Materials. We search for the best hyper-parameter combination for methods based on deep learning without pre-training.
>
> When it comes to methods not grounded in deep learning but specifically tailored for gene representation learning - PCA, Gene2vec, and GIANT - we did not adjust their parameters. However, in light of your feedback, we agree that it is important to consider the hyper-parameters for these methods. We have performed experiments by adjusting the dimensions of the resultant gene embeddings, thereby testing the performance across different models. The search range for these dimensions is set between 32 and 256. For this analysis, we utilized the datasets derived from the heart tissue. Due to the word limit, we recorded the raw score with standard deviation in the pdf file attached to the Author Rebuttal.
>
> Table 1: Min-max scaled benchmarking score table for different optimized models, where the star represents the optimized version.
>
> | Methods | ASW | AUC | iLISI | GC | CGR | NO | Average |
> |-----------------------------|-------------------------------------|--------------------------------------|---------------------------|----------------------------------------|---------------------------------------|--------------------------------------|-----------------------------|
> |PCA*      | 0.17                                | 0.37                                 | 0.76                      | 0.08                                   | 0.47                                  | 0.90                                 | 0.46                        |
> |Gene2vec* | 0.70                                | 0.91                                 | 0.00                      | 0.68                                   | 0.00                                  | 0.32                                 | 0.44                        |
> |GIANT*    | 1.00                                | 0.00                                 | 0.40                      | 0.00                                   | 0.16                                  | 0.00                                 | 0.26                        |
> |MuSe-GNN  | 0.00                                | 1.00                                 | 1.00                      | 1.00                                   | 1.00                                  | 1.00                                 | **0.83**                       |
>
> Based on the results presented in Table 1, we can conclude that even when compared with the other three competitors, which have been optimized for gene representation learning tasks, MuSe-GNN consistently outperformed the other methods.
>
> **2.   Can you confirm that MuSE-GNN has been trained with the same random seeds and thus the results are comparable?**
>
> We appreciate your observation and question. We confirm that the same random seeds were consistently utilized during the training phase for all benchmarking methods and MuSe-GNN. We intend to provide more comprehensive code in the subsequent version of our work.
>
> **3. can you explain better this sentence in the appendix: "Methods such as WSMAE, GAE, VGAE, and MAE, which are based on MLP/GNNs and inspired by other methods, are configured with the best hyperparameters based on parameter searching to ensure fairness."?**
>
> We appreciate your question. We now elaborate our statement below:
>
> The methods under discussion, namely WSMAE, GAE, VGAE, and MAE, are primarily predicated on Multi-Layer Perceptron (MLP) or Graph Neural Networks (GNNs) frameworks and draw inspiration from various other methodologies. To uphold fairness in our study, these methods were fine-tuned and configured with their most conducive hyper-parameters.
>
> The hyper-parameter tuning process entails carrying out multiple experiments with various hyper-parameter configurations to identify the one that delivers the most superior model performance.
>
> However, given that we have now incorporated hyper-parameter tuning even for methods that do not rely on deep learning, we will strive to improve the clarity of this sentence in the forthcoming revision of our manuscript.
>
> **4. Your model uses 349M parameters. How many parameters GIANT uses? If they don't match, can you train a version of MuSE-GNN with a comparable number of parameters to GIANT?**
>
> We appreciate your question. In our experiments, we found that GIANT consumes approximately 260 MB of memory. This result is relatively close to the 349 MB memory usage of our method, suggesting that the memory footprint of both approaches is comparably sized.
>
> Upon further examination of the GIANT model and GIANT paper, we have found that its architecture cannot be modified due to the absence of a user-accessible API for changing the number of layers. We do modify the number of latent dimensions and choose the largest model now. Moreover, we remain open to exploring further possibilities in future research.
>
> **5. Perhaps, as mentioned by the authors, the model is quite "heavy" in terms of number of parameter and one needs to bring down the embedding dimension to avoid OOM errors.**
>
> We appreciate your question and concern. Our ablation tests have indeed shown that excluding raw gene expression from the input significantly impacts MuSe-GNN's performance. Moreover, we have optimized our model to require only a single GPU for training and inference as mentioned in Appendix D.4 of the Supplementary Materials. Nonetheless, we will keep on exploring the optimization of memory efficiency in the future.
>
> Please let us know if you have further questions, and we will do our best to address them. Once again, we thank you for your thoughtful review.

---

> > ### Comment · Reviewer_cB5G · 2023-08-14
> > **Thanks**
> >
> > I have read your rebuttal and appreciated your effort. I now think that this paper is mature enough to make it to the Neurips bar and have changed my score accordingly. Good luck!

---

> > > ### Author Response · Authors · 2023-08-14
> > > **We appreciate your feedback**
> > >
> > > Thank you very much for your recognition and wishes! Your time and efforts are greatly appreciated.

---

### Author Rebuttal · Authors · 2023-08-08

# General response

We would like to thank the reviewers for their overall positive comments on the aims of our manuscript, as well as their insightful comments and inquiries.

We appreciate all the reviewers for their comments about the strengths of MuSe-GNN, including the importantance of the topic (supported by all four reviewers), elegant solutions with informative ablation tests (supported by reviewers cB5G and vvZM), significant improvement over other methods (supported by reviewers anSh and vvZM), meaningful downstream applications (supported by reviewers anSh and vvZM), and a well-written manuscript (supported by reviewers cB5G, mFBG, and vvZM).

We have diligently addressed all the comments, which include clarifying the model's novelty (as raised by reviewers anSh and mFBG), improving the presentation of the model (as suggested by reviewer mFBG), refining our experimental design (as per the feedback from reviewers cB5G and mFBG), and expanding the discussion on related work (as noted by reviewers mFBG and vvZM).

Furthermore, we underscore the importance of our work within the realm of computational biology and provide supplementary materials to advocate for the wider application of AI in scientific research. Once again, we are immensely grateful for the invaluable comments and suggestions from all reviewers.

---

### Decision · Program_Chairs · 2023-09-21

**Decision:**

Accept (poster)

**Comment:**

The paper proposes to integrate multimodal machine learning and deep graph neural networks for learning unified gene representations from multimodal biological graph data. The proposed model, referred to as Multimodal Similarity Learning Graph Neural Network, or MuSe-GNN, aims to learn gene representations containing functional similarity across different contexts in a joint latent space. Experiments are conducted on real-world biological datasets and results show significant improvements over SOTA models.

Reviewers expressed some concerns in their original reviews, regarding (1) potential unfair comparison with baselines due to lack of fine-tuning of their hyperparameters, (2) limited novelty in the proposed method as it combines well-known techniques, (3) inadequate description of the methodology, as well as (4) missing comparison with highly relevant baselines. The authors’ rebuttal adequately addressed all these concerns and clarified some key issues. As a result,  multiple reviewers raised their scores, which reached a consensus in support of the acceptance of the paper.  The authors are encouraged to incorporate all these changes to the final version of the paper.